# The Gestational Pathologies Effect on the Human Milk Redox Homeostasis: A First Step towards Its Definition

**DOI:** 10.3390/nu15214546

**Published:** 2023-10-26

**Authors:** Chiara Peila, Lorenzo Riboldi, Elena Spada, Alessandra Coscia, Ignazio Barbagallo, Giovanni Li Volti, Fabio Galvano, Diego Gazzolo

**Affiliations:** 1Neonatal Unit, Department of Public Health and Pediatrics, University of Turin, 10100 Turin, Italy; 2Department of Biological Chemistry, Medical Chemistry and Molecular Biology, University of Catania, 95131 Catania, Italy; 3Neonatal Intensive Care Unit, University of Chieti-Pescara, 66100 Chieti, Italy

**Keywords:** human milk, breastfeeding, GSH, LOOHs, preeclampsia, gestational diabetes mellitus, newborn nutrition

## Abstract

Background. Human Milk (HM) is a dynamic nourishment; its composition is influenced by several conditions such as gestational age, maternal diet and ethnicity. It appears important to evaluate the impact that gestational pathologies have on HM components and if their presence, as a source of oxidative stress in the mother, influence milk’s redox homeostasis. To assess the effect of Preeclampsia (PE) and Gestational Diabetes Mellitus (GDM) on some aspects of human milk redox homeostasis, we chose to investigate both oxidative and antioxidant aspects, with, respectively, Lipid hydroperoxides (LOOHs) and Glutathione (GSH). Methods. Women with PE, GDM and who were healthy were recruited for this study. Colostrum, transitional and mature milk samples were collected. GSH and LOOHs levels were measured using a spectrophotometric test. To investigate the effect of pathology on redox homeostasis, a mixed linear model with unistructural covariance structure was performed. Results. A total of 120 mothers were recruited. The GSH concentration results were significantly lower in GDM women than in healthy women only in colostrum (*p* < 0.01). No other differences emerged. LOOHs was not detectable in almost all the samples. Discussion. Our study is the first to extensively evaluate these components in the HM of women with these gestational pathologies. The main observation is that GDM can alter the GSH level of HM, mainly in colostrum.

## 1. Introduction

Human milk (HM) is universally recognized as the gold standard of infants nutrition, insomuch as breastfeeding should be considered a health issue and not only a lifestyle choice [1,2]. Besides the nutritional importance of its macronutrients, it is well known that its numerous benefits are mediated by several bioactive substances that can regulate newborns’ growth and development [3,4]. These components can perform multiple functions, playing a protective role from neonatal infections, apart from ensuring the long-term benefits on health and neurodevelopment. HM being a dynamic fluid, its constituents constantly change during lactation phases, according to the time of the day and during a single feeding [5]. In addition to this, its composition is also heavily influenced by other conditions such as gestational age (GA), maternal diet, ethnicity and various diseases [5,6]. The two main gestational pathologies, Gestational diabetes mellitus (GDM) and Preeclampsia (PE), can affect not only lactation but also milk composition.

GDM is a pathological condition defined as any degree of glucose intolerance first recognized during pregnancy [7]. Its incidence depends on diagnostic criteria which are not universally recognized, but its frequency is unanimously increasing in recent years, dependent on women entering pregnancy at an older age and more frequently being obese or overweight [8]. This can determine several delivery complications and constitutes a significant risk factor for newborn morbidity, mainly that the offspring of women with GDM are at an increased long-term risk of type 2 diabetes [9,10]. GDM also holds the ability of delaying lactogenesis’ onset and affecting the composition of human milk throughout its various phases [11].

PE affects about 2–8% of all pregnant women and can be associated with various important maternal and fetal complications [12,13]. In PE, there is an abnormal interaction between placental and maternal tissues with the production of oxidant substances and proinflammatory cytokines, which would be able to damage the vascular endothelium of the mother and generalize the damage [14]. In light of the pathogenetic mechanisms, PE can interfere with the physiological functioning of the mammary glands and consequently alter the production and transfer of nutrients and other components in milk [15].

Bearing these considerations in mind, it appears important to evaluate the impact that these gestational pathologies have on HM composition but, until now, previous studies have focused mainly on the specific biomarkers associated with GDM or PE, often evaluating a single lactating phase or only term delivery [15,16]. It therefore seems clinically appropriate to assess if their presence, as a source of oxidative stress in the mother, can influence milk’s redox homeostasis. Notably, Glutathione (GSH) and Lipid hydroperoxides (LOOHs) appear to have been the target of extensive research, which, however, fails to deliver useful information around their presence both in HM itself and in the HM of pathological mothers. More specifically, GSH has often been used as one of the indicators of total antioxidant capacity in HM, as a substrate of the main antioxidant enzymes [17]. On the other hand, LOOHs are important intermediates of peroxidative reactions induced by reactive species that disrupt membrane structure and function and can be deleterious to cells [18].

In order to assess and evaluate the effect of Preeclampsia and Gestational Diabetes Mellitus on some aspects of human milk’s redox homeostasis, we chose to investigate both oxidative and antioxidant aspects, with, respectively, Lipid hydroperoxides and Glutathione, during all lactation phases. 

## 2. Methods

### 2.1. Setting and Population

This is a longitudinal study with prospective data collection conducted in the Neonatal Unit, Department of Public Health and Pediatrics, University of Turin. Newborns’ mothers were recruited prospectively after delivery at Neonatal Unit. Written informed consent was obtained from all participants and approval from the local ethics committee of the Italian Association of Human Milk Donor Banks (AIBLUD) was obtained.

Given the exploratory nature of our study in the field of the HM redox homeostasis of pathological mothers and the lack of previous studies that could provide assumptions for accurate sample size calculation, it was decided that a minimum of 30 women for each group will be consecutively recruited, based on studies with similar topics [19,20,21].

The women included in the study were divided, subsequently, into three different study groups, according to the following characteristics:

Preeclamptic group: all women who have developed preeclampsia according to the PE definition (Artery blood pressure > 140/90 after 20 weeks of gestational age and proteinuria > 290 mg/L, possibly associated with headache, edema, scotomas and epigastric pain) [22], regardless of maternal age, gestational age at childbirth, parity and twins. 

Gestational diabetes mellitus group: all women who have developed gestational diabetes mellitus according to GDM definition (any degree of glucose intolerance first recognized during pregnancy) [7], regardless of maternal age, gestational age at childbirth, parity and twins.

Healthy women group: the control group was formed at the same time and made up of normotensive and euglycemic women who also met the same exclusion criteria, regardless of maternal age, gestational age at childbirth, parity and twins. 

Criteria exclusion: Women with seizure, psychiatric syndromes, history of alcohol or drug abuse, autoimmune pathologies, chorioamnionitis, evidence of other pregnancy complications like chronic hypertension, type 1 or type 2 diabetes mellitus, renal or liver disease, mastitis and a newborn with congenital anomalies or infection.

Variables collected were maternal data (age, ethnicity, medically assisted procreation, parity, weight gain in pregnancy, smoke, delivery type, twins) and neonatal data (gestational age; gender; birthweight; small for gestational age (defined as birthweight below the 10th percentile); large for gestational age (defined as birthweight higher than the 90th percentile), intrauterine growth restriction).

### 2.2. Collection Human Milk Samples

According to standard criteria, we classified as “colostrum” the milk collected in the first 7 days after delivery; “transition milk”, the milk collected from the 8th day to 14th day after delivery; and “mature milk”, the milk collected after the 15th day [23]. All the breast milk samples were collected using the same procedure outlined below. After washing hands and cleaning the nipple, the women extracted the milk in the presence and under the guidance of a physician. Fresh milk samples were collected in the morning (between 9 a.m. and 12 a.m.) into sterile, disposable, high-density polyethylene sealed bottles (Flormed srl, Naples, Italy). Milk expression was obtained by means of an electric breast pump (Medela Italia srl, Calderara di Reno (BO), Italy) with standard extraction methods. According to current guidelines and to collect full pumping samples, the extraction session was stopped 2 min after the outflow of the last drops of milk.

Milk expression by the other breast was performed only in cases in which it was not possible to obtain a minimum at least of 10 mL from a single breast.

### 2.3. Biochemical Analysis

Whole milk samples were divided into aliquots and stored at −80 °C until analysis. They were then defrosted in a 37 °C bath and immediately analyzed. Regarding GSH levels, the concentrations of non-protein thiol groups (RSH), approximately 90% of the GSH content, in 200 μL of milk were measured using a spectrophotometric test based on the reaction of thiol groups with acid 2,2-dithio-bis-nitrobenzoic (DTNB) at λ = 412 nm (εM = 13,600).

LOOHs levels in milk samples were assessed following oxidation from Fe_2_+ to Fe_3_+ in presence of orange xylenol at λ = 560 nm. One milliliter of the mixture contained 100 μL of milk sample, 100 μM orange xylenol, 250 μM ferrous ammonium sulfate, 90% methanol, 4 mM butylated hydroxytoluene and 25 mM H_2_SO_4_. After 30 min of incubation at room temperature, the absorbance at λ = 560 nm was measured with a Hitachi U2000 spectrophotometer. Calibration was achieved using hydrogen peroxide (0.2–20 μM). The results are expressed in nanomoles (nmol).

### 2.4. Statistical Methods

In the description of the sample, the categorical variables were presented as frequencies (percent), while the continuous variables were presented as mean (standard deviation) or median (interquartile range) (IQR) according to their distribution.

Birthweight was transformed in z-score according to the country-specific Italian Neonatal Study (INeS) charts [24]. Newborns with a birth weight lower than the 10th or higher than the 90th centile were classified as Small for GA (SGA) or Large for GA (LGA), respectively. Regarding GSH, to investigate the distribution by HM phase, a pathology-specific box plot was formed. Then, GSH concentrations were normalized with the more appropriate Box–Cox transformation. To investigate the effect of pathology in the GSH concentration in the 3 different phases, a mixed linear model with an unstructured covariance structure, mother as a random effect and fixed effects was performed. 

The fixed effects were HM phase, pathology, smoker, delivery mode, newborn GA and mother age (continuous), and the interaction between phase hm × pathology. Distributions of GSH were graphically assessed and the transformation regression was used to estimate the Box–Cox power parameter (λ) to normalized GSH distribution. A *p* value < 0.05 was used to define statistical significance. With regard to LOOH concentrations, it was not possible to make an inference analysis, having only the values of 19 samples in 254 samples.

SAS, version 9.4 (SAS Institute Inc., Cary, NC, USA), was used to process data and fit statistical models.

## 3. Results

### 3.1. Demographic Characteristics 

A total of 120 mothers were recruited for our study, divided as follows: 35 in the GDM group, 39 in the PE group and 46 in the healthy group. Table 1 shows the demographic characteristics of mothers and newborns in all three groups. Regarding the maternal characteristics, the median age is similar. The fraction of primigravida in diabetic mothers is lower than in the healthy women group. Moreover, the percentage of cesarean section is high in all groups, as is the percentage of spontaneous pregnancies.

Concerning the neonatal characteristics, the GA is lower in pathology groups than in the healthy group; the GA median in weeks (IQR) is 37 [31–39] in healthy group, while it is 32 [29–35] in the PE group and 36 [33–39] in the GDM group. In the Healthy group, 52% of neonates were born at term (GA ≥ 37 weeks) vs. 21% in the PE group and 46% in the GDM group. Diabetic women have a higher percentage of LGA, while PE women have a higher percentage of IUGR and SGA. In the three groups, the same percentage of twin pregnancies is observed. Furthermore, the mothers of the GDM group and Healthy group delivered both term and preterm newborns, while the group of PE mothers mainly delivered prematurely.

### 3.2. Characteristics of the Human Milk Samples

A total of 254 HM samples were collected. Particularly concerning the Healthy group, a total of 100 samples were collected, of which 36 were colostrum, 32 were transitional milk and 32 were mature milk. In the PE group, a total of 84 samples were collected, of which 31 were colostrum, 30 were transitional milk and 23 were mature milk. In the end, regarding the GDM group, a total of 70 samples were collected, of which 31 were colostrum, 21 were transitional milk and 18 were mature milk.

### 3.3. GSH 

GHS has been detected in all types of HM, regardless of the lactation phase, gestational pathologies and the GA at childbirth. 

Figure 1 shows the box-plot GSH distribution by HM phase and pathology (PE, GDM and Healthy). The box-plot by HM phase and pathology (GDM vs. Healthy) shows a positively skewed GSH distribution at each HM phase. In the colostrum of the GDM group, that the 25% of the measures results equal to 0, λ = 0.25 is the more appropriate Box–Cox transformation. The GSH concentration in the GDM group is significantly lower in colostrum than in transitional milk (*p* = 0.02) and mature milk (*p* < 0.01), while no significant difference was detected between transitional and mature milk. Regarding the interaction between groups, the GSH concentration results were significantly lower in GDM women than in healthy women only in colostrum (*p* < 0.01). No other differences emerged.

### 3.4. LOOHs

LOOHs were not detectable in most samples (235 of 254 samples).

Table 2 shows in detail the description of the samples in which LOOHs were found and their relative concentrations.

In the HM of mothers that gave birth at term of GA, belonging to the Healthy group, the LOOHs were found only in 6 samples out of 49, and the mean concentration was 1.05 nmol. The levels (nmol) detected in the different samples were, respectively, 0.88 and 1.22 in 2 out of 17 colostrum samples, 0.55 and 1.44 in 2 out of 15 transitional milk samples and 12.11 and 0.44 in 2 out of 17 mature milk samples. In the HM of mothers that gave birth preterm, belonging to the Healthy group, the LOOHs were found only in 1 sample out of 51 and the concentration was 0.55 nmol in a mature milk sample. In the PE group, the LOOHs were found in 1 sample out of 84 and the concentration was 34.22 nmol in the transitional milk sample. Regarding the GDM group, the LOOHs were found only in 11 samples out of 70, and the mean concentration was 4.55 nmol. The levels (nmol) detected in the different samples were, respectively, 0.88, 1.77, 2.33 and 4.33 in 4 out of 31 colostrum samples, 31.22, 1.88 and 1.88 in 3 out of 21 transitional milk samples and 1.88, 1.55, 1.88 and 0.55 in 4 out of 18 mature milk samples.

## 4. Discussion

There is evidence that newborns are already more vulnerable to oxidative stress in the first days of life, due to the inefficiency of their antioxidant defense system and the increase in free radical production outside the womb. Neonates need more antioxidants from the first phase of breast milk to protect their health [25,26]. However, HM remains the first choice for the nutrition of term and preterm infants due to its demonstrated short and long term effects and benefits, regardless of maternal gestational pathology [1,2,3,4]. The previous literature shows how GDM and PE negatively influence the lactogenesis and physiological maturation of breast milk by modifying the general content of biologically active molecules [15,16]. Moreover, exposure to maternal pathology can influence the antioxidant burden of colostrum and mature milk and can adversely affect the infant’s health, playing a role in the pathogenesis of various conditions such as necrotizing enterocolitis, bronchopulmonary dysplasia and retinopathy of prematurity, among others [27]. Therefore, it is important to know the effects of the single gestational pathology on the composition of a mother’s own milk and, further still, on its redox homeostasis, in order to adequately modify nutrition or any supplements especially in the case of premature infants. To the best of our knowledge, our study is the first that investigates and compares some aspects of the HM redox homeostasis of healthy and pathological mothers, assessing both oxidative and antioxidant aspects with, respectively, Lipid hydroperoxides (LOOHs) and Glutathione (GSH). 

Regarding the characteristics of our populations, the descriptive analysis shows that the GA is lower in pathology groups than in the Healthy group. In particular, only 21% of neonates were born at term in the PE group. Knowing that HM composition is gestational-age-dependent, GA was included as a covariate in the analysis model, obtaining results “adjusted” for GA estimates.

GSH is considered one of the main water-soluble antioxidants in biological fluids [25]. Previous research has already investigated GSH levels, specifically in the HM of healthy women [26,28]. More precisely, the available data provide indications regarding the total antioxidant capacity of HM and the enzymes that use glutathione as a substrate, which are increased in the colostrum of healthy full-term mothers, probably to address the greater vulnerability of newborns to oxidative stress and relative immaturity of their endogenous antioxidant means [26]. 

While the increase in GSH concentration appears to be a universal cellular response to oxidative stress, some diseases, like GDM, seem to be linked to a decrease in its levels in the body [29]. In the presence of metabolic deregulation and uncontrolled hyperglycemia, oxidative stress and the formation of ROS rise significantly. This ultimately leads to an increase in lipid peroxidation and its byproducts and a depletion in antioxidants such as GSH [29,30]. It also has been certified that, in diabetic subjects, the increase in lipid peroxidation and the depletion in antioxidants, in addition to the insulin-dependent expression of enzymes degrading GSH, contribute to decreasing its concentration in the body [30]. As far as GSH and PE are concerned, numerous previous studies have investigated the overall maternal redox status related to gestational hypertension and the authors generally agree in stating that there is a significant correlation between oxidative stress and pregnancy pathology. Especially in PE, the levels of oxidative stress markers increase and the concentrations of GSH and other antioxidant molecules in plasma decrease [31]. Some studies have demonstrated this association even before the diagnosis of PE, demonstrating that high blood levels of oxidative stress markers (e.g., malondialdehyde), and low levels of glutathione and other antioxidants before the 20th week of gestation, correlate with a higher probability of subsequently diagnosing the presence of preeclampsia [32,33,34,35]. 

Our study is the first that evaluates the GSH concentrations during all lactation phases of mothers with GDM and PE. Regarding data on the GSH values of diabetic mothers, our findings seem to match previous studies highlighting diabetes’ higher oxidative burden; the GSH concentration results were significantly lower in colostrum in the GDM group than in the Healthy group. Concerning the HM of the PE group, our study demonstrated that there are no differences in the GSH concentrations compared to the healthy group, highlighting how the oxidative maternal status, in this case, does not directly affect the milk concentrations. 

The LOOH assay is very delicate. Though the method used in our study for LOOHs measurement is the most used due to its sensitivity in biological samples, [36] and though, to avoid further oxidation during the measurement phase, we added butylated hydroxytoluene to the reaction solution to stabilize the chemical solvents and increase the sensitivity of the sample measurement, LOOHs are detectable in only 19 samples of the 254 total HM samples tested. It is interesting to note how LOOHs were detected in very few milk samples without an apparent correlation with pathology, gestational age, type of birth or lactation phase. However, there are no studies in the literature that have directly investigated the presence of lipid hydroperoxides in the human milk of PE or GDM women with which to compare our results. On the other hand, our study is in partial disagreement with the results of the previous data in the field of oxidative stress in the milk of hypertensive women. In fact, previous studies report, in the milk of women with hypertension, lower values of total antioxidant capacity and vitamins A and E and higher values of oxidative stress index and total peroxides, malondialdehyde (MDA), conjugated dienes and bases of Schiff (some lipid peroxidation products) [37,38]. These results are in agreement with the pathogenetic mechanism of the disease and with the data concerning the plasma levels of these molecules that, in hypertensive and preeclamptic women, are higher than the levels in healthy women [35,39,40,41,42]. Only one, however, reports higher levels of polyphenols in the hypertensive group and, simultaneously, lower levels of MDA; in this case, the authors hypothesize a mechanism of adaptation of the maternal organism to protect the newborn from oxidative stress [43]. 

In general, moreover, in the literature, there are few studies that have evaluated the presence of these molecules in HM. Most of them are focused on the evaluation of the oxidative status after the various processing methods used in the storage of HM, and conflicting results have been reported concerning the occurrence of lipid peroxidation and the production of dangerous final products, probably because of the variability of the molecules investigated (such as malondialdehyde and 4-hydroxy-nonenal) and the different methodologies employed [20,44,45,46]. There are also other studies that have evaluated the differences in the HM oxidative status related to different maternal and neonatal factors [21,46,47,48]. Our study deviates from the previous ones in the choice of molecules to evaluate lipid peroxidation (mainly malondialdehyde and 4-hydroxy-trans-2-nonenal) and also because the maternal pathologies are not taken into consideration into consideration in the analysis.

A first limitation of our study is that we have detected the presence of LOOHs only in a few samples, although the technique used to analyze LOOHs is the most sensitive according to the literature and the sample size seemed adequate according to previous studies focused on HM redox homeostasis. Given this observation, we can assume that this specific molecule, when investigated alone, does not fully reflect the lipo-peroxidative status of breast milk. However, knowing that LOOHs are produced through the lipid peroxidation of HM PUFA, it can be hypothesized that the gestational pathology does not seem to have an effect in increasing the peroxidation activity related to LOOH production. In future studies, to obtain a wider and critical picture of the effects of the gestational pathologies on HM oxidative status, it would be interesting to expand not only the sample size but also the lipid peroxidation products. In association with these, it would also be interesting to evaluate in parallel the content of PUFA and FFA, knowing that lipid peroxidation is a process under which oxidants, such as free radicals, attack lipids containing carbon–carbon double bonds.

A second limitation of our study is that we could not measure, in the different HM samples, other antioxidant molecules and the potential confounding factors of the redox homeostasis represented by ferrous iron, Vitamins E, C and A, L-Argininine, iodine, zinc, selenium and copper. In fact, for example, iodine has been shown to be directly involved in the regulation of HM oxidative stress, correlating negatively with superoxide dismutase, catalase and glutathione peroxidase activities [49,50,51]. Moreover, vitamins C and E have been shown to reduce the concentration of oxidative products in maternal plasma, and the frequency of preeclampsia in a high-risk population and the concentration of each vitamin in colostrum is positively correlated with its respective concentration in plasma [38,52,53]. Bearing these considerations in mind, our results and their interpretation are limited and future studies should simultaneously investigate different antioxidant substances together with the several possible potential confounding factors of this complex system to try and define the antioxidant capacity of HM. Although a reliable assessment of the confounding factors of HM redox homeostasis is very difficult, a practical example is represented by vitamins due to their large inter-subject variation in the concentrations related to differences in lifestyles, nutritional intake, food fortification and supplement use, and also due to some methodological factors, such as the day of collection or time elapsed from previous breastfeeding and the previous meal.

In the end, this is an explorative study in which the sample size was defined on previous similar studies. It is possible that the observed non-differences are due to low power and the absence of true equality between the comparison groups. This study may be useful for planning future studies with a more precise clinical question (non-inferiority, equality, superiority design) and with a power appropriate for the purpose.

## 5. Conclusions

In conclusion, this study is a first step towards the definition of the Human Milk Redox status of women with gestational pathologies that potentially affect human milk composition and redox homeostasis itself. The main observation is that GDM can alter the GSH concentration of human milk mainly in colostrum, more than throughout other lactating phases. On the contrary, in case of PE, regarding the molecules investigated, the composition of HM milk is preserved. Further studies are needed to confirm our findings and to identify the potential correlations between the composition of HM and the gestational pathologies. These future findings are important to individualize and modify maternal therapies and supplement the nutrition of the preterm newborns.

## Figures and Tables

**Figure 1 nutrients-15-04546-f001:**
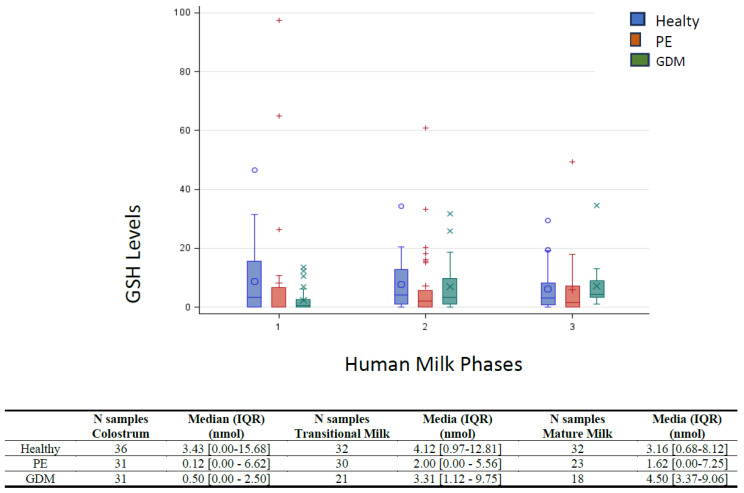
Box-plot of GSH by lactation stage and pathology. Median and interquartile range (IQR) were reported for each lactation stage (1 = Colostrum; 2 = Transitional milk; 3 = Mature milk). A *p* < 0.05 was used to define statistical significance. Preeclamptic group (PE); Gestational diabetes mellitus group (GDM); Healthy women group (Healthy).

**Table 1 nutrients-15-04546-t001:** Basal characteristics of mothers and newborns.

		Healthy N = 46	PE N = 39	GDM N = 35
Maternal characteristics				
Age (years)	median (IQR)	33.5 [31–37]	35 [31–38]	33 [30–36]
Italian	n (%)	35 (76.1)	31 (79.5)	21 (60.0)
Cesarean Section	n (%)	25 (54.4)	28 (71.8)	18 (51.4)
Spontaneous Pregnancy	n (%)	39 (84.8)	31 (79.5)	33 (94.3)
Weight gain (kg)	mean (SD)	10.9 (4.75)	10.4 (5.65)	9.2 (5.38)
Primigravida	n (%)	29 (63.0)	25 (64.0)	14 (40.0)
Smoker	n (%)	6 (13.0)	2 (5.1)	11 (31.4)
Newborn characteristics				
Singleton	n (%)	38 (82.6)	36 (92.3)	31 (88.6)
IUGR	n (%)	2 (4.4)	16 (41.0)	7 (20.0)
GA (weeks)	median (IQR)	37 [31;39]	32 [29–35]	36 [33–39]
Girl	n (%)	19 (41.3)	19 (48.7)	21 (60.0)
Birthweight (g)	mean (SD)	2345 (1028)	1542 (720)	2540 (1035)
Birthweight (z-score)	mean (SD)	−0.21 (0.934)	−1.16 (0.810)	−0.06 (1.406)
SGA	n (%)	6 (13.0)	19 (50.0)	8 (22.9)
LGA	n (%)	2 (4.4)	0 (0.0)	8 (22.9)

**Table 2 nutrients-15-04546-t002:** Values of LOOHs samples if detectable.

	All Samples		Colostrum		Transitional Milk		Mature Milk	
	N.	Mean (nmol)	N.	Mean (nmol)	N.	Mean (nmol)	N.	Mean (nmol)
Healty	7/100	1.05	2/36	0.88–1.22	2/33	0.55–1.44	3/31	4.36
PE	1/84	34.22	0/31	-	1/30	34.22	0/23	-
GDM	11/70	4.55	4/31	2.32	3/21	11.66	4/18	1.46

## Data Availability

The data presented in this study are available on request from the corresponding author.

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
