# Peer review of "The Gestational Pathologies Effect on the Human Milk Redox Homeostasis: A First Step towards Its Definition"

_nutrients, 2023, doi:10.3390/nu15214546_

Round 1

Reviewer 1 Report (Previous Reviewer 2)

Comments and Suggestions for Authors

The paper of Dr. Chiara Peila, Dr. Lorenzo Riboldi, and co-authors has been improved for publication in Nutrients.

Please read carefully the paper to avoid spelling mistakes like carachteristics instead of characteristic in the following paragraph:

‘Regarding the carachteristics of our populations, the descriptive analysis shows that

the GA is lower in pathology groups than in Healthy group. In particular, only 21% of

neonates were born at term in the PE group. Knowing that HM composition is gestational

age-dependent, GA was included as covariates in the analysis model obtaining results

"adjusted" for GA estimates’.

Comments on the Quality of English Language

The English Style is correct.

Author Response

We want to thank the reviewer for their kind comments and useful suggestions that have been taken into account in the present revised form.

We checked the paper and the typing errors have been corrected in the text as suggested.

Reviewer 2 Report (Previous Reviewer 1)

Comments and Suggestions for Authors

Dear Authors, 

   Thank you very much for addressing all my comments.

Author Response

We want to thank the reviewer for their kind comments and useful suggestions that have been taken into account in the present revised form.

Reviewer 3 Report (New Reviewer)

Comments and Suggestions for Authors

This article, titled "Human Milk Redox Homeostasis: the Effect of Gestational Pathologies" (Nutrients 2673096), is submitted for the "Pediatric Nutrition" section of the Special Issue titled "Human Milk Composition and Children's Nutrition: Latest Advances and Prospects."

The objective of this study was to evaluate the impact of Preeclampsia (PE) and Gestational Diabetes Mellitus (GDM) on the total oxidative load of breast milk. We have chosen to investigate both oxidative and antioxidant aspects, specifically Lipid Hydroperoxides (LOOHs) and Glutathione (GSH), respectively.

This intriguing work has several aspects that I believe should be considered:

1.    In the abstract, the methods section should include a description of the study design.

2.    In the methodology section, the study's design should be clearly stated, along with the characteristics of the study population. Ethical committee approval should be mentioned, as well as details regarding the calculation of the sample size and the sampling method employed. Additionally, in the statistical methods section, it should be indicated how the normality of the distribution was assessed and specify the software used for data analysis.

3.    In the results section, there are highlighted yellow lines, suggesting that this might be a resubmission. In Table 1, a comparison should be made between patients with PE and healthy subjects, patients with GDM and healthy subjects, and a comparison between PE and GDM should also be included.

4.    In line four of the results section, it is mentioned that "le da" (gives) is similar. This is not correct; it should be statistically assessed using the appropriate tests for parametric and non-parametric distribution of variables. Therefore, I believe that Table 1 should be further elaborated.

5.    In the discussion section, it is crucial to acknowledge the study's limitations for future research.

6.    The conclusions should align with the objectives and results presented.

I believe that the text should be revised as there appear to be some errors and omissions that need attention.

Comments on the Quality of English Language

This article, titled "Human Milk Redox Homeostasis: the Effect of Gestational Pathologies" (Nutrients 2673096), is submitted for the "Pediatric Nutrition" section of the Special Issue titled "Human Milk Composition and Children's Nutrition: Latest Advances and Prospects."

The objective of this study was to evaluate the impact of Preeclampsia (PE) and Gestational Diabetes Mellitus (GDM) on the total oxidative load of breast milk. We have chosen to investigate both oxidative and antioxidant aspects, specifically Lipid Hydroperoxides (LOOHs) and Glutathione (GSH), respectively.

This intriguing work has several aspects that I believe should be considered:

1.    In the abstract, the methods section should include a description of the study design.

2.    In the methodology section, the study's design should be clearly stated, along with the characteristics of the study population. Ethical committee approval should be mentioned, as well as details regarding the calculation of the sample size and the sampling method employed. Additionally, in the statistical methods section, it should be indicated how the normality of the distribution was assessed and specify the software used for data analysis.

3.    In the results section, there are highlighted yellow lines, suggesting that this might be a resubmission. In Table 1, a comparison should be made between patients with PE and healthy subjects, patients with GDM and healthy subjects, and a comparison between PE and GDM should also be included.

4.    In line four of the results section, it is mentioned that "le da" (gives) is similar. This is not correct; it should be statistically assessed using the appropriate tests for parametric and non-parametric distribution of variables. Therefore, I believe that Table 1 should be further elaborated.

5.    In the discussion section, it is crucial to acknowledge the study's limitations for future research.

6.    The conclusions should align with the objectives and results presented.

I believe that the text should be revised as there appear to be some errors and omissions that need attention.

Author Response

We want to thank the reviewer for their kind comments and useful suggestions that have been taken into account in the present revised form.

  1. In the abstract, the methods section should include a description of the study design.

In the methods section of the abstract, the study design part has been better defined as suggested, with the limit of 200 words wanted by Nutrients.

  1. In the methodology section, the study's design should be clearly stated, along with the characteristics of the study population.Ethical committee approval should be mentioned, as well as details regarding the calculation of the sample size and the sampling method employed. Additionally, in the statistical methods section, it should be indicated how the normality of the distribution was assessed and specify the software used for data analysis.

We have modified the methods section as required, adding the software used for data analysis and defining more in detail the sampling method employed and the characteristics of the study population. Regarding the sample size, it was assessed a-priori on the basis of the number of sample studies previously conducted by experts in the same field (i.e. doi:10.1111/jpc.13676; doi:10.1080/08035250410022495; https://doi.org/10.3390/antiox11081472; doi:10.1016/j.clinbiochem.2005.05.004). The approval of the ethics committee was mentioned in the text and the approval codes are reported in the specific section as requested from Nutrients format. Moreover, we added the methods to assess normality distribution for GSH and to estimate the best box-cox transformation.

  1. In the results section, there are highlighted yellow lines, suggesting that this might be a resubmission. In Table 1, a comparison should be made between patients with PE and healthy subjects, patients with GDM and healthy subjects, and a comparison between PE and GDM should also be included. In line four of the results section, it is mentioned that "le da" (gives) is similar. This is not correct; it should be statistically assessed using the appropriate tests for parametric and non-parametric distribution of variables. Therefore, I believe that Table 1 should be further elaborated.

The highlighted yellow lines have been removed.

Regarding the lack of statistical analysis for the characteristics of the study groups, our team has decided to follow the guidelines provided by “STROBE Statement: Strengthening the Reporting of Observational Studies in Epidemiology” (doi:10.7326/0003-4819-147-8-200710160-00010-w1. PMID: 17938389.).

Regarding Table 1 and the descriptive characteristics, STROBE Statement recommends: “Inferential measures such as standard errors and confidence intervals should not be used to describe the variability of characteristics, and significance tests should be avoided in descriptive tables.”

For this reason, we decided not to apply p values in Table 1 that is a table with the sole purpose of describing how the variables are distributed in the three groups under study.

Given the observational nature of the study presented, some variables are expected to have a different distribution. Such differences, although they are awaited, are beyond the scope of our study to demonstrate them. The model for the outcome analysis was chosen on the basis of the variables associated with the outcome itself, as already known and was corrected for these variables. Furthermore, we describe 14 variables (only in table 1) and, choosing alpha=.05 for each test, the whole type 1 error is very high (51%). For these reasons we believe that comparison tests can complicate the study by introducing the problem of multiple tests and therefore increasing the whole risk of type I error and creating confusion about the aim of the study.

  1. In the discussion section, it is crucial to acknowledge the study's limitations for future research.

The limitations of this study have been extended in the discussion paragraph as required.

  1. The conclusions should align with the objectives and results presented.

The conclusion section has been modified as suggested.

Round 2

Reviewer 3 Report (New Reviewer)

Comments and Suggestions for Authors

I have carefully reviewed the article with the incorporated modifications, as well as the authors' response letter to the comments made. I have some suggestions; the authors have not taken into account the suggestion to compare the results of baseline characteristics in the various groups in Table 1. However, this is essential to consider adjustments in the subsequent model that will be conducted. Therefore, I still believe that it is necessary to apply comparison tests among the three studied groups in Table 1.

In the limitations section, it is necessary to mention that the small sample size, and the model conducted a sample size calculation for this study, which poses a risk of type two error. Hence, future studies with larger sample sizes are required.

Comments on the Quality of English Language

I would also like to suggest to the authors that they review the English.

Author Response

I have carefully reviewed the article with the incorporated modifications, as well as the authors' response letter to the comments made. I have some suggestions; the authors have not taken into account the suggestion to compare the results of baseline characteristics in the various groups in Table 1. However, this is essential to consider adjustments in the subsequent model that will be conducted. Therefore, I still believe that it is necessary to apply comparison tests among the three studied groups in Table 1.

Regarding the Table 1 and the statistical model subsequently used, we preferred a different approach, than the one suggested by the reviewer for choosing the model. To estimate the association between pathology and GSH concentration, we used the approach of causal relation. In essence, we selected the variables whose association with outcome and/or exposure is known, or highly suspected, and we describe the association among all these variables. In this way, it was possible to select the minimum set of variables useful for estimating the effect of pathology (exposure) on GSH concentration (outcome). The advantage of this approach is that the model is not sample-dependent, is generalizable, and no over-correction is introduced. (doi: 10.1111/resp.12238; doi:10.7326/0003-4819-147-8-200710160-00010-w1.)

The approach of performing a univariate pre-screening test on all covariates, suggested by the reviewer, has some criticisms. The main one is that it fails to identify residual confounding, and, in addition, the resulting model is sample dependent. (doi: 10.1007/s12630-017-0833-0)

In the limitations section, it is necessary to mention that the small sample size, and the model conducted a sample size calculation for this study, which poses a risk of type two error. Hence, future studies with larger sample sizes are required.

Thank you for the suggestion, we have expanded the limitation paragraph as requested.

This manuscript is a resubmission of an earlier submission. The following is a list of the peer review reports and author responses from that submission.

Round 1

Reviewer 1 Report

Comments and Suggestions for Authors

Dear Authors,

   This paper offers a very innovative approach to understand the complex mechanisms involved in oxidative stress regulation in human milk, particularly in case of such prevalent pathologies as Preeclampsia or gestational diabetes.

This work has valuable elements to consider:

n  Human milk has been collected and compared at different phases of lactation (colostrum, transitional and mature milk).

n  The statistical analysis has dealt with a complex mixed linear model.

n  Figure 1 is very illustrative.

n  In general, the manuscript is very well written, and the discussion contains relevant comments to put these findings in context.

However, there are some minor aspects to precise/clarify:

a.      The expression “gestational diseases” is correct but, to me, is too much similar to “gestational trophoblastic disease (GTD)” which is an entity far different from those conditions studied here. I suggest that the term “gestational diseases” may be replaced by “Obstetric pathologies”.

b.      Results, page 3, line 140: median age is similar. It would be more precise to say that median age does not show significant differences between groups.

c.      Table 1 should include a new column with statistical significance (p value) as well as the statistical test used for comparisons.

d.      Page 4, point 3.3 GSH: the information contained in the first paragraph already appears in Figure 1. Please, try to avoid duplicities.

e.      Discussion: I miss a short comment about the role of some nutrients (e.g., iodine) as potential confounders in the redox homeostasis of human milk.

Author Response

Dear Sir / Madam,

enclosed please find the revised version of the MS entitled “Human milk redox homeostasis: the gestational pathologies effect” by Peila C.; Riboldi L. et al. We want to thank you and the reviewer for their kind comments and useful suggestions that have been taken into account in the present revised form. In particular:

We trust that now, the MS in the present revised version will meet the criteria for publication on Nutrients.

Best regards

a. The expression “gestational diseases” is correct but, to me, is too much similar to “gestational trophoblastic disease (GTD)” which is an entity far different from those conditions studied here. I suggest that the term “gestational diseases” may be replaced by “Obstetric pathologies”.

As suggested by the reviewer, we modified all “gestational diseases” with “gestational pathologies” to make the term less similar to “gestational trophoblastic disease (GTD)”

b. Results, page 3, line 140: median age is similar. It would be more precise to say that median age does not show significant differences between groups.

c. Table 1 should include a new column with statistical significance (p value) as well as the statistical test used for comparisons.

Table 1 is a table that has the sole purpose of describing how the variables are distributed in the three groups under study. Given the observational nature of the study presented, some variables are expected to have a different distribution. Such differences, although they are awaited, are beyond the scope of our study to demonstrate them. The model for the outcome analysis was chosen on the basis of the variables associated with the outcome itself, as already known and was corrected for these variables. For this reason we believe that comparison tests can complicate the study by introducing the problem of multiple tests and therefore increasing the risk of type I error.

d. Page 4, point 3.3 GSH: the information contained in the first paragraph already appears in Figure 1. Please, try to avoid duplicities. 

The Result section has been shortened according to reviewer’s suggestion.

e. Discussion: I miss a short comment about the role of some nutrients (e.g., iodine) as potential confounders in the redox homeostasis of human milk.

Discussion section has been expanded according to reviewer’s suggestion.

Reviewer 2 Report

Comments and Suggestions for Authors

The paper of Chiara Peila, Lorenzo Riboldi, and collaborators address parameters of interest to evaluate human breast milk quality. In the field, we need reports like this one, sharing data connected to Gestational diabetes mellitus and Preeclampsia, as an international effort to evaluate preventative strategies for the mother’s health and the consequences on the baby's future health.

Evaluating the redox potential of milk collected from 3 groups of mothers (healthy, Gestational Diabetes Mellitus, Preeclampsia). It is relevant to the field of Breast milk analysis. Breast milk is a complex food. Consequently, I think that the paper is correctly directed to Nutrients. However, the paper can be read to be more in line with a clinician's paper.

The paper provides data on colostrum and on later lactation stage. The parameters measured are classic and properly discussed, measurements on breast milk are new.

Minor alterations.

I advise the authors to re-read their captions of illustrations and tables.

(1) Headings of Table-2. Please correct « Tansitional » with « Transitional ».

Comments on the Quality of English Language

 Minor editing of English language required

Author Response

Dear Sir / Madam,

enclosed please find the revised version of the MS entitled “Human milk redox homeostasis: the gestational pathologies effect” by Peila C.; Riboldi L. et al. We want to thank you and the reviewer for their kind comments and useful suggestions that have been taken into account in the present revised form. In particular:

We trust that now, the MS in the present revised version will meet the criteria for publication on Nutrients.

Best regards

However, the paper can be read to be more in line with a clinician's paper.

We modified the manuscript as suggested and we trust the revised form is more useful for clinicians.

I advise the authors to re-read their captions of illustrations and tables.

(1) Headings of Table-2. Please correct « Tansitional » with « Transitional ».

Typing errors have been corrected in the text as suggested.  

Reviewer 3 Report

Comments and Suggestions for Authors

The main problem concerns the lack of statistical analysis for the characteristics of the study groups. According to the data provided by Authors for PE group the gestational age seems significantly lower. For this reason, groups that differ significantly in terms of week of gestation should not be compared, since milk composition is gestational age-dependent. Taking into account the above details of statistical analysis must be provided in the manuscript. Without these key details the presented results raise doubts and do not fully reflect the true relationships. Moreover, the statistical details are missing in Table 2 and Figure 1.

Abstract

Line 19-20 „.GSH and LOOHs were quantified using an ELISA test.”

From the description in the method section, it appears that the methods without antibodies were used. This point needs detailed clarification.

Line 25-26

“The main observation is that GDM can alter the antioxidant composition of HM mainly in colostrum, but in case of PE, the composition of HM milk is preserved.”

This point needs verification. The study was not focused on antioxidant composition of human milk, but is limited to the two factors only.

Line 93-94

The approval number of the ethics committee was not provided in the manuscript

Line 136

“3.1. Demographic characteristics and Table 1 – statistical analysis is missing

In the section concerning “Newborn characteristics” for pregnancy complicated by PE significantly lower gestational age translates to differences in Birth weight (g) and Birth weight (z-score). Moreover, there is a well-known fact that the quantitative composition of breast milk depends on the week of delivery. For this reason, groups that differ significantly in terms of week of gestation should not be compared, because the obtained results will not indicate true relationships.

Line 176 The table header, legend and specific values for the statistical significance coefficient p are missing.

Discussion

Line 216-217

“In order to complete the literature gap, we chose to investigate not only all the different types of HM (colostrum, transitional milk and mature milk) but also the HM composition of mothers who have given birth at term and preterm.”

There is no adequate commentary in this regard. The results are not included in the work.

Line 252-264

The authors do not refer critically to their own results, which are inconsistent with the data presented by other research teams. In addition, there are no references to the works of other authors in this field (line 262-264).

Line 288-294

“It will also be important to analyze the effect on HM of:

 ›different types of diabetes (GDM, Type 1 Diabetes and Type 2 Diabetes)

 ›different types of Hypertension (Chronic Hypertension, Gestational Hypertension, PE and HELLP syndrome).

 ›different drugs used and their potential interaction on the different HM biological components.”

This part of the discussion is puzzling.

Line 295-298 Relevant references are missing

“Finally, considering the rising prevalence of type 2 diabetes in association with obesity, we believe it is important to plan future studies to determine possible effects on human milk in this population. These future findings are important to individualize and 298 modify the maternal therapies and supplement the nutrition of the preterm newborns.”

There is a wealth of literature available in this area that should be cited in this passage.

Conclusions

Line 303-307

“On the contrary, in case of PE, the composition of HM milk is preserved”

Based on the results presented in the paper, such a statement raises doubts, especially when the relevant parameters of statistical analysis are not presented.

Comments on the Quality of English Language

Terms that need correction

Line 67-68 “HM of pathological mothers”

Line 177 Figure 1 Box-Plot of GSH by HM phase and pathology

In this case, a better term is “lactation stage or milk type”

Line 183 “Tansitional milk” – should be “Transitional milk”

“n.” – should be n/N

Line 214 “the HM redox homeostasis of healthy and pathological mother,”

Author Response

Dear Sir / Madam,

enclosed please find the revised version of the MS entitled “Human milk redox homeostasis: the gestational pathologies effect” by Peila C.; Riboldi L. et al. We want to thank you and the reviewer for their kind comments and useful suggestions that have been taken into account in the present revised form. In particular:

We trust that now, the MS in the present revised version will meet the criteria for publication on Nutrients.

Best regards

The main problem concerns the lack of statistical analysis for the characteristics of the study groups. According to the data provided by Authors for PE group the gestational age seems significantly lower. For this reason, groups that differ significantly in terms of week of gestation should not be compared, since milk composition is gestational age-dependent. Taking into account the above details of statistical analysis must be provided in the manuscript. Without these key details the presented results raise doubts and do not fully reflect the true relationships. Moreover, the statistical details are missing in Table 2 and Figure 1.

Table 1 is a table that has the sole purpose of describing how the variables are distributed in the three groups under study. Given the observational nature of the study presented, some variables are expected to have a different distribution. Such differences, although they are awaited, are beyond the scope of our study to demonstrate them. The model for the outcome analysis was chosen on the basis of the variables associated with the outcome itself, as already known and was corrected for these variables. For this reason we believe that comparison tests can complicate the study by introducing the problem of multiple tests and therefore increasing the risk of type I error.

Table 2 (now table 3) is also uniquely descriptive of the few LOOH values found in the milk samples tested. Given the small numbers, a statistical analysis was not carried out to compare the concentrations between the different groups because it was not considered appropriate (for the PE group, LOOH was detectable in a single sample).

Abstract

  • Line 19-20 GSH and LOOHs were quantified using an ELISA test.” From the description in the method section, it appears that the methods without antibodies were used. This point needs detailed clarification.

The paragraph has been modified according to reviewer’s suggestion. 

  • Line 25-26 “The main observation is that GDM can alter the antioxidant composition of HM mainly in colostrum, but in case of PE, the composition of HM milk is preserved.”This point needs verification. The study was not focused on antioxidant composition of human milk, but is limited to the two factors only.

The claim has been changed according to reviewer’s suggestion.

  • Line 93-94 The approval number of the ethics committee was not provided in the manuscript

The approval number has been added at the end of the manuscript in the section “Institutional Review Board Statement”

  • Line 136 3.1. Demographic characteristics and Table 1 – statistical analysis is missing

In the section concerning “Newborn characteristics” for pregnancy complicated by PE significantly lower gestational age translates to differences in Birth weight (g) and Birth weight (z-score). Moreover, there is a well-known fact that the quantitative composition of breast milk depends on the week of delivery. For this reason, groups that differ significantly in terms of week of gestation should not be compared, because the obtained results will not indicate true relationships.

Statistical analysis section has been evaluated according to reviewer’ suggestions and supervised by our Epidemiology & Statistics service. 

Table 1 is a table that has the sole purpose of describing how the variables are distributed in the three groups under study. Given the observational nature of the study presented, some variables are expected to have a different distribution. Such differences, although they are awaited, are beyond the scope of our study to demonstrate them. The model for the outcome analysis was chosen on the basis of the variables associated with the outcome itself, as already known and was corrected for these variables. For this reason we believe that comparison tests can complicate the study by introducing the problem of multiple tests and therefore increasing the risk of type I error.

  • Line 176 The table header, legend and specific values for the statistical significance coefficient p are missing.

The Figure 1 and Table 2 header has been changed as requested.

Discussion

  • Line 216-217 “In order to complete the literature gap, we chose to investigate not only all the different types of HM (colostrum, transitional milk and mature milk) but also the HM composition of mothers who have given birth at term and preterm.” There is no adequate commentary in this regard. The results are not included in the work.

The text has been changed as requested.

  • Line 252-264 The authors do not refer critically to their own results, which are inconsistent with the data presented by other research teams. In addition, there are no references to the works of other authors in this field (line 262-264).

We modified the paragraph, and we added references as suggested.

  • Line 288-294 “It will also be important to analyze the effect on HM of: ›different types of diabetes (GDM, Type 1 Diabetes and Type 2 Diabetes) ›different types of Hypertension (Chronic Hypertension, Gestational Hypertension, PE and HELLP syndrome). ›different drugs used and their potential interaction on the different HM biological components.” This part of the discussion is puzzling. Line 295-298 Relevant references are missing “Finally, considering the rising prevalence of type 2 diabetes in association with obesity, we believe it is important to plan future studies to determine possible effects on human milk in this population. These future findings are important to individualize and 298 modify the maternal therapies and supplement the nutrition of the preterm newborns.” There is a wealth of literature available in this area that should be cited in this passage.

The text has been changed as requested.

Conclusions

  • Line 303-307 “On the contrary, in case of PE, the composition of HM milk is preserved” Based on the results presented in the paper, such a statement raises doubts, especially when the relevant parameters of statistical analysis are not presented.

We modified the paragraph as suggested and we trust the revised wording is more accurate.

Terms that need correction

  • Line 67-68 “HM of pathological mothers”
  • Line 177 Figure 1 Box-Plot of GSH by HM phase and pathology
  • In this case, a better term is “lactation stage or milk type”
  • Line 183 “Tansitional milk” – should be “Transitional milk”
  • “n.” – should be n/N
  • Line 214 “the HM redox homeostasis of healthy and pathological mother,”

The text has been changed and typing errors have been corrected in the text as suggested.